# Apoptosis Regulation in Osteoarthritis and the Influence of Lipid Interactions

**DOI:** 10.3390/ijms241713028

**Published:** 2023-08-22

**Authors:** Frederike Werry, Emilia Mazur, Lars F. H. Theyse, Frank Edlich

**Affiliations:** 1Institute of Biochemistry, Faculty of Veterinary Medicine, University of Leipzig, 04103 Leipzig, Germany; frederike.werry@vmf.uni-leipzig.de; 2Soft Tissue & Orthopaedic Surgery Service, Department for Small Animals, College of Veterinary Medicine, University of Leipzig, 04103 Leipzig, Germany; emilia.mazur@kleintierklinik.uni-leipzig.de

**Keywords:** osteoarthritis, apoptosis, lipid interactions, BCL-2 protein family, chondrocyte, autophagy, cartilage, macrophages, synovial fibroblasts, MOMP

## Abstract

Osteoarthritis (OA) is one of the most common chronic diseases in human and animal joints. The joints undergo several morphological and histological changes during the development of radiographically visible osteoarthritis. The most discussed changes include synovial inflammation, the massive destruction of articular cartilage and ongoing joint destruction accompanied by massive joint pain in the later stadium. Either the increased apoptosis of chondrocytes or the insufficient apoptosis of inflammatory macrophages and synovial fibroblasts are likely to underly this process. In this review, we discuss the current state of research on the pathogenesis of OA with special regard to the involvement of apoptosis.

## 1. Osteoarthritis

Osteoarthritis (OA) is a debilitating joint disease affecting both humans and companion and farm animals [1,2]. Diarthrodial or synovial joints are essential for the articulation between opposing parts of the skeleton. The joints between long bones typically consist of two epiphyses covered with articular hyaline cartilage, a synovial joint sac and a joint capsule providing stability. The articular hyaline cartilage consists of chondrocytes within an extracellular matrix (ECM). The ECM is produced and maintained by the chondrocytes. The ECM is composed of collagen type II fibers and glycosaminoglycans (GAG). Glycosaminoglycans can be subdivided into two major constituents: hyaluronic acid and aggrecan. Hyaluronic acid is an extremely long unbranched polysaccharide, whereas aggrecan consists of a protein core with numerous short, unbranched sulphated mucopolysaccharides, including chondroitin 4-sulphate, chondroitin 6-sulphate and keratan sulphate. The aggrecan complex is connected to the hyaluronic acid by its linking proteins. The collagen type II fibers in the ECM are anchored within the subchondral chondral bone. The crosslinking between the collagen fibers and glycosaminoglycans is responsible for the rigidity of the cartilage matrix. The chondrocytes have a flat appearance near the joint surface and a more rounded appearance in the deeper layers of the cartilage. Within the ECM, chondrocytes are found as individual cells or as isogenic groups surrounded by a dense matrix known as capsule. Chondrocytes almost completely depend on diffusion from the synovial fluid to cover their metabolic requirements. The synovial fluid can be characterized as a plasma ultrafiltrate with a supplemental high amount of hyaluronic acid. In contrast to hyaline cartilage, the subchondral bone, synovial membrane and joint capsule have a direct vascular supply. The synovial membrane is lined with synoviocytes consisting of type A and type B synovial cells. The type A or macrophagic cells predominantly function as scavenger cells. The type B or fibroblast-like cells are responsible for the production of the synovial fluid, including hyaluronic acid and lubricin. Hyaluronic acid acts as a lubricant and forms an equilibrium with the hyaluronic acid in the ECM. Lubricin is a surface-active glycoprotein providing boundary lubrication and preventing cell and protein adhesion. This means that chondrocytes rely on a delicate anabolic and katabolic balance to maintain joint homeostasis [3]. Joint autophagy plays an important role in the orderly degradation and recycling of cellular and ECM components within the joint [4,5]. The enhancement of joint autophagy has been described as an early sign of OA [6]. Although joints can withstand large loading forces, the regeneration potential is very limited once damage does occur. The ensuing cascade of joint alterations, including responses within the hyaline cartilage and responses related to the synovial membrane, subchondral bone, non-weightbearing bone surfaces within the joint sac and joint capsule, is known as OA. Any disturbance of the joint equilibrium can initiate an osteoarthritic response, and chondrocyte apoptosis is part of this process [7]. The classical division between primary OA and secondary OA has been supplemented with several subtypes including metabolic disease, mechanical disbalance and repetitive strain [8]. In the concept of primary OA, the assumption is that there is no initial causative trauma to the joint, and OA develops spontaneously and shows slow progression. In the concept of secondary OA, the assumption is that there is an initiating trauma with the development and progression of OA as a sequel. The concept of early OA has been postulated to be able to detect OA before irreversible damage to the joint has been established. This would open up a window for regenerative intervention to stop the progression of OA [9]. Emerging concepts to assess the different stages and progression of OA are based on the description of clinical phenotypes and molecular endotypes [1,2]. In the early stages, OA is characterized by damage to the ECM and depletion of chondrocytes [10]. The resulting ECM debris will initiate a scavenging response of the type A synoviocytes and a clear inflammatory response of the synovial membrane. The cascade of cytokine and adipokine driven inflammation will also exert its effect on the composition and amount of the synovial fluid produced by the type B synoviocytes. In addition, the chronic inflammatory reaction results in the thickening of the joint capsule, subchondral bone remodeling and chondrophyte and osteophyte formation, with pain and loss of joint function as an end result. For a long time, cartilage degeneration, chondrocyte depletion and chondrocyte apoptosis have been placed in the center of OA research [10]. Ideally, the joint should be approached as an organ, including all of the previously described joint structures. Rheumatoid arthritis and OA share many of the disease mechanisms leading to the loss of joint integrity [11,12]. In rheumatoid arthritis, the emphasis is on the derailment of the synoviocytes with cartilage degeneration as a sequel [13,14]. The disturbance of the homeostasis of cell apoptosis plays a critical role in the initiation and progression of both OA and rheumatoid arthritis and is a key research target.

## 2. Regulated Cell Death in the Pathogenesis of Osteoarthritis

OA shows the progressive destruction and reduction of articular cartilage through the failure of the extracellular matrix. These transformations of the matrix are probably conferred by chondrocytes as opposed to the “dead” matrix. Anabolic activity, phenotypic stability and the survival of chondrocytes are extremely important for the homeostasis and functionality of articular cartilage. Unsurprisingly, proliferation activity is markedly reduced in chondrocytes from osteoarthritic tissue [15]. The low proliferation activity of chondrocytes is probably due to an avascular extracellular matrix that cannot compensate for cell loss and the absence of a germ cell layer (Figure 1). The regression of the cartilage could therefore result from the reduced cell proliferation of the chondrocytes. However, lacunar emptying as a second typical feature of osteoarthritis is a strong indicator that cell death programs are involved in the development of cartilage degeneration in osteoarthritic joints [16,17]. At least in part, this cell death is apoptosis [18]. Apoptotic rates of chondrocytes, ranging from less than 1% to up to 20%, have been observed in the cartilage of osteoarthritic joints [15,19].

During apoptosis, cells manage their orderly demise to keep the multicellular organism functional [20]. Beyond maintaining entire organism homeostasis, hematopoiesis and development, a central function of apoptosis is to protect against tumorigenesis [21]. Apoptotic cells detach from their cell-cell contacts, round up and form apoptotic bodies, presenting an intact plasma membrane with phosphatidylserine (PS) acting as the signal for phagocytosis, usually without an inflammatory response [22]. The apoptotic pathways are mediated by cysteinyl aspartate proteases (Caspases). Apoptosis can be triggered by external signals (Figure 2; left panel).

Then, the adapter protein Fas-associated death domain (FADD) associates with activated members of the tumor necrosis factor (TNF) receptor family, including Fas and TNF receptor-1 [23]. FADD contains the death domain for receptor interaction and the death effector domain, which is exposed through this interaction. The binding of pro-caspase-8 to the death effector domain forms the death signaling complex (DISC), which promotes the autoactivation of this initiator caspase [24]. Caspase-8-dependent cleavage of the effector procaspases activates caspase-3, -6 or -7, and in turn, promotes apoptosis through the cleavage of cellular substrates [25].

After a cell stress which probably underlies osteoarthritis, a different apoptosis program is initiated, dependent on the BCL-2 protein family (Figure 2; right panel). Mitochondrial apoptosis occurs through a different initiator caspase following the permeabilization of the outer mitochondrial membrane (OMM). The BCL-2 protein family splits into three branches: pro-survival proteins, pro-apoptotic proteins and BH3-only pro-apoptotic proteins. BCL-2, among others, is a pro-survival protein of the mitochondrial signaling pathway [26]. The apoptosis-promoting proteins are the BCL-2-associated X protein (BAX), BCL-2 antagonist killer 1 (BAK) and perhaps BCL-2-related ovarian killer protein (BOK). Hitherto, it has not been possible to show the origin of the opposing mechanisms of action compared to the pro-survival proteins, despite structural similarities [27,28]. However, BAX and BAK commit the cell to apoptosis by the permeabilization of the OMM following their activation. The BH3-only proteins, including BIM, BID, BAD, NOXA and several others, have the short BH3 sequence as the only conserved motif domain of the BCL-2 protein structure, as well as the function of signaling cell stress to the BCL-2 proteins on the OMM [29]. These signals can either be external stress stimuli (e.g., radiation, treatment with chemotherapeutic drugs, etc.) or internal stimuli such as oxidative stress, high cytosolic concentrations of Ca^2+^ or genetic damage [30]. The various stress stimuli that lead to BAX/BAK activation and the form that the active BAX complex takes remain unclear despite a quarter of a century of ongoing research. Nevertheless, there are some insights into the sophisticated mitochondrial apoptosis signaling chain (Figure 3). 

The continuous retrotranslocation of BAX and BAK from the mitochondria to the cytosol is required for cell survival and appears to be the major activity of pro-survival BCL-2 proteins [31,32]. The two independent processes of OMM association and the activation of BAX or BAK are required for apoptosis, with the activation of BAX depending on the dwell time on the OMM [33]. OMM permeabilization results in the release of cytochrome *c* (cyt *c*) and other intermembrane space (IMS) proteins into the cytoplasm [34]. Cyt *c* binds the cytosolic apoptosis protease activating factor-1 (APAF-1), complexing to the formation of a heptameric protein ring called apoptosome. The complex recruits the initiator pro-caspase-9 to the caspase recruitment domain (CARD), followed by autoactivation. This in turn leads to the activation of the effector caspases-3, -6 and -7, followed by the cleavage of cellular substrates [35,36]. With the activation of the caspases-3, -6 and -7, the extrinsic and intrinsic pathways merge into each other. In many cells, procaspase-8-cleavage can activate the BH3-only protein BID. Once activated, the C-terminal truncated BID shifts to the mitochondria and activates caspase-9 as the intrinsic pathway is often required for a cell to die after an extrinsic signal [37]. Because of this central role of BID, a special mechanism has been developed to inhibit truncated BID by hexokinase-dependent retrotranslocation from the OMM [38]. BID has also been suggested to oligomerize [39] and perhaps permeabilize the OMM itself [40], adding further potential complexity to apoptosis signaling.

## 3. Apoptosis and Inflammation in Osteoarthritis

Chronic OA develops over a long period of time, suggesting limited cell death rather than an onset of chondrocyte apoptosis. Because some observed features in osteoarthritic cartilage do not appear to be typical of apoptosis, the terms “chondroptosis” and “paraptosis” were introduced [41]. In this way, for example, the lack of nuclear fragmentation observed after treatment with the insulin-like growth factor 1 (IGF-1) is highlighted. In this context, it has also been shown that the expression of the IGF-1 receptor is increased [42]. It should be considered that nuclear fragmentation perhaps occurs with different kinetics in different cell types. The apparent decreased phagocytosis of apoptotic bodies in OA may be due to the composition of the extracellular matrix of cartilage. The restriction of the phagocytosis of apoptotic remnants by other cells leads to secondary necrosis and probably to a mixture of cell death that could be mistaken for a specific type of program. In chondrocytes, initially the amount of Golgi and endoplasmic reticulum (ER) increases noticeably, which could suggest rising rates of protein synthesis und secretion. Furthermore, numerous autophagic vacuoles are found. The lack of proper apoptosis progression may also lead to another atypical feature frequently observed in OA: the cells show a massive extrusion of cell contents into the empty lacunae and the surrounding matrix structures. Some cells appear to disintegrate into fragments and vesicles [43]. These remnants could be apoptotic bodies, as they are separated from dying cells [44]. When phagocytosis is disturbed, the cellular remnants have three different fates. First, cellular material is digested within the ER compartments, followed by autophagy, and finally, cellular debris is released into the empty lacunae [43]. Multiple components of the cartilage, e.g., collagen, proteoglycans or hyaluronan, are damaged due to the oxidative stress induced by reactive oxygen species (ROS) [45,46]. As the mitochondrial electron transport chain is an important source for ROS, mitochondrial dysfunction following apoptotic OMM permeabilization seems to play a central role in this aspect of the pathogenesis of OA [47,48]. Further support comes from other changes in mitochondria, e.g., modified mitochondrial respiratory chain activity, ATP production, and mitochondrial membrane potential (ΔΨm). The contribution of inflammatory cytokines such as IL-1β or TNF-α is also discussed [49,50]. 

Different pathways lead to inflammation in OA. In the IL-1β signaling pathway, interleukin-1β triggers chondrocyte apoptosis as an important inflammatory mediator and therefore appears to play an essential role in the pathogenesis of OA [51]. The overexpression of IL-1β can be found in the cartilage and synovial tissue. At the same time, the expression of the IL-1 receptor antagonist (IL-1Rα) in chondrocytes seems significantly reduced [52,53]. Nitric Oxide (NO) seems to be an important player during OA development. It is suggested that NO mediates a reduced synthesis of IL-1Rα in the chondrocytes caused by IL-1β [54]. In addition, the TNF-family can initiate extrinsic apoptosis. The protein family contains several cytokines produced by macrophages, monocytes, lymphoid cells and fibroblasts, among others. They transmit their cellular responses in immunity, inflammation, cell differentiation, proliferation and apoptosis through 29 receptors of the TNFR-family [55,56]. TNFR1 contains the previously mentioned death domain and induces apoptosis after binding TNF-α through the activation of NFκB and JNK MAPKs. The activation of the pro-inflammatory transcription factor, nuclear factor kappa B (NFκB), has the consequence that cytokines, including interleukin 1-β and TNF-α, are again expressed as positive feedback [57]. C-Jun N-terminal kinases (JNKs) and p38s represent subgroups of mitogen-activated protein kinases (MAPKs) activated by cellular stress and inflammatory cytokines [58]. Besides apoptosis, this pathway is also involved in cell growth, differentiation and transformation [59]. The phosphorylation of c-Jun, a part of the transcription factor AP-1, by JNKs results in an interaction with other transcription factors to form the complete “activator protein 1” (AP-1). The formation leads to the expression of proteins, which have importance for the Fas/FasL apoptosis signaling pathway [60,61]. On the one hand, the phosphorylation of pro-apoptotic proteins, such as p53, BAD, BimEL and BimL, by 2-JNKs leads to the truncation of BID and therefore the activation of BAX and BAK. In addition, p53 can actively intervene in the apoptotic process by upregulating the pro-apoptotic genes, such as PUMA (p53 upregulated modulator of apoptosis) [62]. Unlike in healthy cartilage tissue, increased rates of phosphorylated JNK and p38 MAPK were detected in OA cartilage, which were activated by the cytokines TNF-α and IL-1. Some studies suggest inhibiting JNK MAPKs as a potential therapeutic tool in rheumatoid arthritis (RA) treatment, leading to the consideration of a possibly similar effect in OA. Recent investigations with animal models of OA showed reduced cartilage degeneration and pain after the inhibition of p38 [63,64,65]. Further, NO production seems to mediate chondrocyte apoptosis through the mitochondria-dependent pathway, and again to increase the synthesis of proinflammatory cytokines, such as IL-1 [66]. In an experimental canine OA model, the selective inhibition of the inducible nitric oxide synthase (iNOS) resulted in the macroscopically visible reduction of cartilage damage. However, NO alone failed to induce apoptosis in cultured chondrocytes [67]. On the other hand, cells tend to undergo apoptosis at low ROS concentrations, whereas high concentrations of ROS and associated low levels of NO appear to promote necrosis [68]. 

An early ER response in chondrocytes is corroborated by the upregulation of the chaperones 78-kDa glucose-regulated protein (Grp78) and BCL-2-associated athanogene-1 (bag-1) in the articular cartilage of OA patients [69]. The ER serves to fold and compose the membrane and secreted proteins before they are transported to other cellular compartments. Ongoing ER stress results in the unfolded protein response (UPR) to minimize the damaging influences [70]. Through the transcriptional activation of the C/EBP homologues protein (Chop) gene, BCL-2 proteins, such as BCL-2 and BIM, are activated [71]. ER stress in human OA cartilage is thought to activate the apoptotic pathways in chondrocytes, leading to a potential role for ER stress in the pathogenesis of OA [72]. Adult cartilage as an avascular tissue makes chondrocytes sensitive for potential oxygen or nutrient depletion, which can lead to hypoxia and catabolic stress and, therefore, ER stress [73]. 

There are, as stated, several autophagic components in the process of chondrocyte degeneration. Among many differences, autophagy shares some common features with apoptosis, such as the absence of inflammation or ATP consumption. In autophagic processes, the fragmentation of the cytoplasm and intracellular components results in the formation of characteristic vacuoles, which are gradually degraded by lysosomes (Figure 4). In the early stages of OA, the chondrocytes show an increased expression of apoptotic, such as caspase-3, as well as autophagic markers, such as microtubule protein 1 light chain 3 (LC3), in the superficial cartilage zone. As degeneration progresses, both markers are present in the chondrocytes of the superficial and middle zones and have been associated with defects in the cartilage repair response, which may be due to the increased activation of death signals and catabolic mechanisms. However, in the deep zones of degenerative cartilage, no autophagic markers but only apoptotic markers were detected at high levels [74]. There are considerations that lysosomal proteins are released into the cytosol during apoptosis and autophagy, similar to mitochondrial proteins [75,76,77]. Caspase-8 has been shown to release cathepsin B, a lysosomal cysteine protease, through lysosomal membrane permeabilization (LMP) [76]. However, the precise mechanism underlying LMP remains to be elucidated. For example, LMP could also be mediated by tBID, which has been observed to translocate to lysosomes during apoptosis [78]. BAX has additionally been suggested to interact with lysosomes during Stauroporine-induced apoptosis [79]. Accordingly, the OA release of lysosomal proteins could result from apoptosis signaling. Since both pathways are interconnected, simultaneous apoptosis and autophagy at different stages of OA are unsurprising. Consequently, the decreased expression of autophagic markers in induced OA is accompanied by an increased level of the apoptosis marker PARP p85 [80]. Further studies need to investigate potential therapeutic approaches targeting OA development. 

The pan-caspase inhibitor Z-VAD-FMK and the caspase-3 inhibitor Z-DEVD-FMK have been shown to suppress chondrocyte apoptosis, which was induced by various medical treatments [81]. An anterior cruciate ligament transaction (ACLT) led to the development of OA in dogs and was therefore used as an animal model. Z-DEVD-FMK, as well as the caspase-9 inhibitor Z-LEHD-FMK, reduced apoptosis in this model [82]. In ACLT analysis in rabbits, an intraarticular injection of Z-VAD-FMK has led to significantly reduced cartilage degradation due to suppressed caspase activity [83]. Thus, the inhibition of caspases holds promise in providing the basis for effective AO therapy. Nevertheless, AO remains a multifactorial disease with a great need for the development of therapeutic strategies in which apoptosis inhibition may be an important factor.

The investigation of further cells in the complex structure of the joint has so far mainly taken place in the process of chronic inflammatory synovitis in the context of RA. This is dominated by the presence of synoviocytes, which can be divided into type A (macrophages) and type B (synovial fibroblasts) synoviocytes (Figure 1). A local proliferation of synovial fibroblasts is quite conceivable [84], whereas macrophages are enriched from the peripheral blood and do not proliferate in the joint. Thus, one must consider the insufficient apoptosis of these cell types and consequently their accumulation for the development of RA and possibly OA. The latter must become the focus of interest in future investigations. In RA, FLIP, a regulator of caspase-8, is highly expressed in macrophages, and studies have shown increased Fas-mediated apoptosis after FLIP restriction [85,86]. On the one hand, FLIP suppression represents a possible therapeutic approach for RA, but on the other hand, this mechanism is quite conceivable in osteoarthritic joints as well. Not only in macrophages, but also partly in synovial fibroblasts, an enhanced activation of NFκB and consequently reduced apoptosis in synovial tissue was observed in RA patients. Its inactivation resulted in the caspase-dependent apoptosis of the macrophages [87]. Another important pathway for macrophages in patients with RA seems to be the phosphatidyl-inositol 3-kinase (PI3K) pathway, protecting cells from apoptosis. With the consideration of reduced apoptosis in joint macrophages, whether in RA or OA, the PI3K-AKT1 pathway should therefore be considered and further investigated as a possible target site for therapeutic intervention during the course of the disease. Besides macrophages, synovial fibroblasts play a major role in the pathogenesis of RA. A reduced apoptosis rate is therefore proclaimed as the main feature and should also be investigated with regard to the development of OA. For example, in the synovial fibroblasts of RA joints, inactive somatic mutations of the tumor-suppressor protein p53, a major player in BAX/BAK activation, have been found [88]. Cells with reduced or lacking p53-dependent apoptosis signaling have a higher threshold for apoptotic stimuli [89,90]. Accordingly, a therapeutic strategy for RA might be to restore sensitivity to or the activity of p53.

As OA is a degenerative disease, a potential role of cell senescent should also be considered. To this end, senescent fibroblast-like synoviocytes have been found to promote OA progression in osteoarthritic joints [91]. Further, synovial fibroblasts, synovial macrophages, osteoblasts and adipocytes seem to be involved in the production of “senescent-associated secretory phenotype (SASP) factors”, which were originally found in chondrocytes [92,93]. Those SAPSs include inflammatory cytokines, immune modulators and proteases. In particular, inflammatory cytokines are the likely culprits for the induction of apoptosis in OA [94].

## 4. Lipids Potentially Modulate Cell Degeneration in Osteoarthritis

Based on the available studies on OA, a central role of apoptosis is likely in both the molecular basis of cell degeneration and the observed inflammatory processes. When considering the underlying molecular processes, the interaction between BAX and BAK with the lipids of the OMM are of paramount importance. The first lipid to consider is cardiolipin (CL), which is thought to directly bind and in fact recruit BAX to the OMM [95]. CL might also be involved in OMM permeabilization by BAX and BAK [96,97]. The polar and negatively charged diphosphatidylglycerol lipid is mainly localized in the inner mitochondrial membrane (IMM) and to a much lesser extent in the OMM [98]. There is a consideration that cardiolipin may as well recruit and activate BID and thus induce tBID-dependent apoptosis through BAX activation [99]. By overexpressing the phospholipid scramblase 3 (PLS-3), CL is increasingly shifted from the IMM to the OMM and consequently an increased apoptosis could be detected [100]. As with CL, a direct interaction of BAX and BAK with cholesterol has not yet been demonstrated, but is suggested. The depletion of cholesterol seems to impair OMM permeabilization [101]. While cholesterol depletion appears to have no impact on the membrane integration of BAX [101], its lateral translocation to lipid microdomains could be reduced [102]. Sphingolipids have been hypothesized as possible additional players in apoptosis regulation. The emphasis is on the pore-forming effect of ceramides, the basic structural elements of all sphingolipids, in the OMM. Additionally, ceramides may interact with BAX in a direct manner to induce their pore-forming activity [103,104]. Similar effects were observed with the transient expression of acid sphingomyelinase, which catalyzes the conversion of sphingomyelin to ceramide. Referring to the BAK structure, there is the consideration of the presence of a phospholipid-binding site that supports the oligomerization of BAK dimers [105]. Besides the membrane-bound lipids, which have been discussed so far, there are also considerations about a direct interaction of BAX with the hydrophobic prostaglandins. The hydrophobic molecules are known to play a key role in the mechanism of inflammatory response [106]. Although apoptosis is usually accompanied without an inflammatory response, an inflammatory component has been demonstrated in the development of OA, so an influence of prostaglandins on BAX activation is quite possible. Recent studies revealed an increase in BAX-dependent cell death after the microinjection of intracellular prostaglandine E2 (PGE2), which seems to induce conformational changes of BAX and lead to insertion in isolated mitochondria [107]. Similar effects have been observed with prostaglandine A2 (PGA2) [108]. However, BCL-2 protein-independent effects are also discussed [109]. Certainly, further investigations on the potential therapeutic benefits of interfering with lipid interactions with BAX and BAK in the context of OA are needed.

## 5. Conclusions

The research on OA has revealed a major contribution of apoptosis to pathogenesis. With the increasing number of authentic animal models, the evidence of apoptotic cell death in OA is growing. However, the research also indicates a complex interplay between different stress response programs and various cell fates among the different types of involved cells, making the breakdown of molecular underpinnings a difficult but needed endeavor. Therefore, a central avenue of future research should be the connection to RA. Especially with regard to RA, investigations into therapeutic approaches have been extensively advanced in recent years. Such findings are also desirable for the therapy of OA, in order to be able to reduce or even remove a significant limitation in everyday life for the corresponding patients.

## Figures and Tables

**Figure 1 ijms-24-13028-f001:**
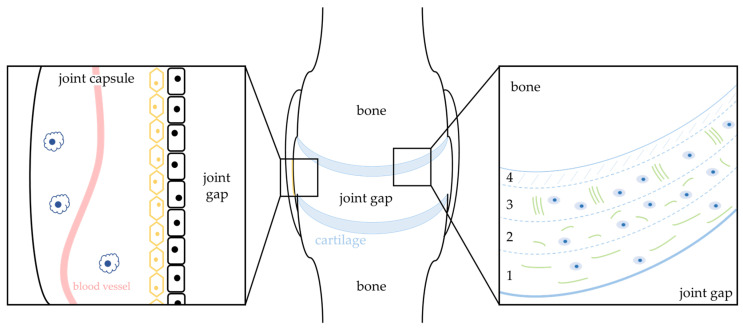
For the pathogenesis of OA, a reduced cell proliferation of chondrocytes is hypothesized (**right box**; blue cells). Apoptotic and autophagic markers are present in the superficial (1) and middle cartilage zones (2). In the deep zones (3) of degenerative cartilage, only apoptotic markers are present. The avascularity of the boundary layer (4) is thought to be the cause of the low proliferation rate of chondrocytes. Collagen and proteoglycans (green) are damaged in the pathogenesis of OA by ROS. Insufficient apoptosis and local accumulation of type A (macrophages; **left box**; dark blue and black cells) and type B (synovial fibroblasts; yellow cells) synoviocytes as in the pathogenesis of RA must be considered for OA as well.

**Figure 2 ijms-24-13028-f002:**
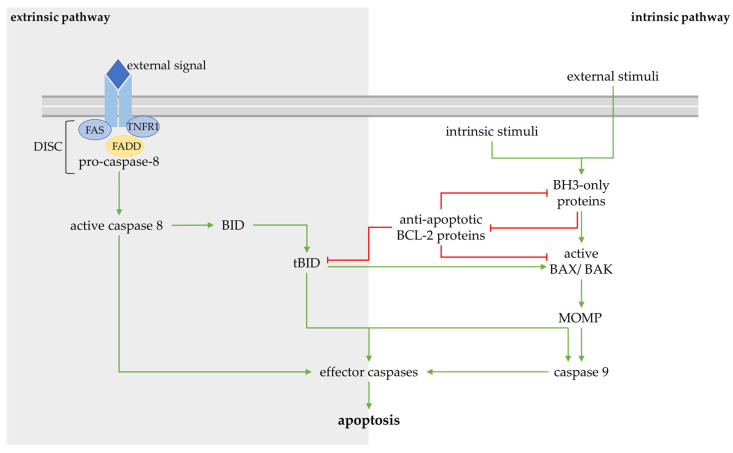
Apoptosis is induced by various external (extrinsic pathway; **left panel**) or internal (intrinsic pathway; **right panel**) stimuli. The extrinsic pathway is mediated via ligands of the death receptors (blue and yellow). After activation (green arrow) of a caspase-cascade, apoptosis is initiated. The intrinsic pathway, initiated by various stimuli, is dependent on interaction with BH3-only proteins. They activate monomeric BAX/ BAK and lead to permeabilization of the outer mitochondrial membrane (MOMP). They inactivate (red arrow) anti-apoptotic BCL-2 proteins. This interplay finally leads to apoptosis again. tBID appears to have a key role affecting both pathways.

**Figure 3 ijms-24-13028-f003:**
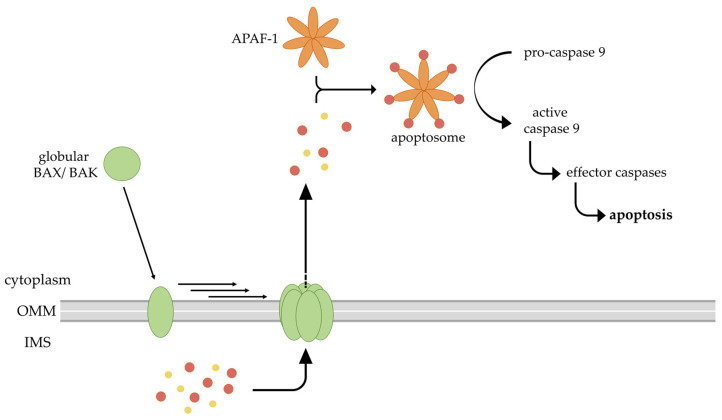
Translocation of monomeric BAX/ BAK (green) to the outer mitochondrial membrane (OMM) and oligomerization leads to the release of intermembrane space (IMS) proteins, such as cytochrome *c* (red) or SMAC (yellow), into the cytoplasm. Cyt *c* binds to APAF-1 (orange) and forms the heptameric apoptosome. The caspase recruitment domain of the apoptosome initiates the autoactivation of caspase 9 and results in a caspase cascade (black arrows). At this point of no return, the cell undergoes apoptosis.

**Figure 4 ijms-24-13028-f004:**
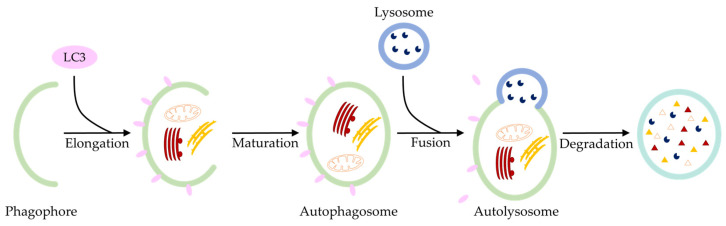
Fragmentation of the cytosplasm and intracellular components (orange and red) in autophagic processes. Characteristic vacuoles, called autophagosomes (green), are degraded by lysosoms (blue) without inflammation or ATP consumption. The autophagic marker LC3 (pink) is detectable in the superficial and middle zones of osteoarthritic joint cartilage.

## Data Availability

No new data were created.

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
