# Peer review of "Apoptosis Regulation in Osteoarthritis and the Influence of Lipid Interactions"

_ijms, 2023, doi:10.3390/ijms241713028_

Round 1

Reviewer 1 Report

This is a basic review of apoptosis in the pathogenesis of osteoarthritis. The manscript is clear and well written , but does not deepend in aspects associated with cell death of potential significance in the pathogenesis of this degenerative disease. In particular, I miss comments about lysosomes including the potential role of Bak/Bax in the permeabilization of the lysosomal membrane. Additionally, the implication of cell senescence, which very often accompanies cell death in most degerative diseases is neglected.

I will recommend  to reduced the text concerning very basic and well-kown  aspects of apoptosis and to expand other points such as those mentiones above.

Author Response

We truly appreciate the constructive, positive, and thoughtful reviewer input.

Concerns

1) The reviewer suggests to include comments about lysosomes including the potential role of Bak/Bax in the permeabilization of the lysosomal membrane.

In the revised manuscript, we have tightened the description of general apoptosis to make room for the description of the relationships between apoptosis signals and lysosome permeabilization on page 7 lines 256-273. We thus follow the suggestion and thank the reviewer for this important point.

2) The reviewer would like to include the implication of cell senescence.

We thank in the reviewer for this important suggestion and discuss the implication of cell senescence in the manuscript on page 8 lines 321-328.

Reviewer 2 Report

The article presented by Werry and colleagues comprehensively describes the involvement of the apoptotic mechanism in the onset and evolution of osteoarthritis, responding in an exhaustive manner to the main topic of the research. Although the topic is interesting and relevant, it would be useful to include a paragraph regarding the therapies used to date and their mechanism of action vs apoptosis, to complete the paper. Compared with other reviews published on the topic, the detailed description of the molecules involved in the apoptotic mechanism and specifically in osteoarthritis is appreciable.

It is definitely necessary to correct some of the references that state “Error! Reference source not found; left panel" as well as pay more attention to these details before submitting an article.

The proposed images are clear and explanatory.

Author Response

We are grateful for the time and effort the reviewer invested in our manuscript and thank him/her for the constructive thoughts.

Concerns

1) The reviewer asks about therapeutic approaches in relation to observations of the role of apoptosis.

We thank the reviewer for bringing up this important point and discuss current therapeutic approaches based on altered apoptosis signaling in the manuscript on page 7 lines 286-293 and page 8 lines 316-320.

2) The reviewer noted some problems with the format of references.

We have retrospectively noticed that when submitting the manuscript, some formatting was destroyed due to the conversion of the files. We will pay special attention to this when resubmitting and apologize for the problems this caused.

Round 2

Reviewer 1 Report

Sorry to say that changes introduced in the revised manuscript are not fully satisfactory.

I send the author s couple of references that can be of help to address the my concerns about the original maniscript

Targeting regulated chondrocyte death in osteoarthritis therapy.

Zhu R, Wang Y, Ouyang Z, Hao W, Zhou F, Lin Y, Cheng Y, Zhou R, Hu W. Biochem Pharmacol. 2023 Jul 26;215:115707. doi: 10.1016/j.bcp.2023.115707.

The role of apoptosis in the pathogenesis of osteoarthritis.

Xiao SQ, Cheng M, Wang L, Cao J, Fang L, Zhou XP, He XJ, Hu YF. Int Orthop. 2023 Aug;47(8):1895-1919. doi: 10.1007/s00264-023-05847-1.

Autophagic cell death is dependent on lysosomal membrane permeability through Bax and Bak.

Karch J, Schips TG, Maliken BD, Brody MJ, Sargent MA, Kanisicak O, Molkentin JD. Elife. 2017 Nov 17;6:e30543. doi: 10.7554/eLife.30543.

Author Response

We sincerely thank the reviewer for the time and effort invested. However, we must refrain from including the proposed work because proposals 1 and 2 do not contain original discoveries. Instead of proposal number 3, we cite the original discovery of BAX-mediated lysosomal membrane permeabilization by Kågedal et al, 2005.